# Predicting Cardiovascular Risk in Athletes: Resampling Improves Classification Performance

**DOI:** 10.3390/ijerph17217923

**Published:** 2020-10-28

**Authors:** Davide Barbieri, Nitesh Chawla, Luciana Zaccagni, Tonći Grgurinović, Jelena Šarac, Miran Čoklo, Saša Missoni

**Affiliations:** 1Department of Biomedical and Specialty Surgical Sciences, Faculty of Medicine, Pharmacy and Prevention, University of Ferrara, 44121 Ferrara, Italy; davide.barbieri@unife.it; 2Interdisciplinary Center for Network Science and Applications, University of Notre Dame, Notre Dame, IN 46556, USA; nchawla@nd.edu; 3Biomedical Sport Studies Center, University of Ferrara, 44123 Ferrara, Italy; 4Polyclinic for Occupational Health and Sports of Zagreb Sports Association with Laboratory of Medical Biochemistry, 10000 Zagreb, Croatia; toncimd@gmail.com; 5Centre for Applied Bioanthropology, Institute for Anthropological Research, 10000 Zagreb, Croatia; jelena.sarac@inantro.hr (J.Š.); miran.coklo@inantro.hr (M.Č.); 6Institute for Anthropological Research, 10000 Zagreb, Croatia; sasa.missoni@inantro.hr; 7School of Medicine, Josip Juraj Strossmayer University of Osijek, 31000 Osijek, Croatia

**Keywords:** medical diagnostic, decision tree, logistic regression, machine learning

## Abstract

Cardiovascular diseases are the main cause of death worldwide. The aim of the present study is to verify the performances of a data mining methodology in the evaluation of cardiovascular risk in athletes, and whether the results may be used to support clinical decision making. Anthropometric (height and weight), demographic (age and sex) and biomedical (blood pressure and pulse rate) data of 26,002 athletes were collected in 2012 during routine sport medical examinations, which included electrocardiography at rest. Subjects were involved in competitive sport practice, for which medical clearance was needed. Outcomes were negative for the largest majority, as expected in an active population. Resampling was applied to balance positive/negative class ratio. A decision tree and logistic regression were used to classify individuals as either at risk or not. The receiver operating characteristic curve was used to assess classification performances. Data mining and resampling improved cardiovascular risk assessment in terms of increased area under the curve. The proposed methodology can be effectively applied to biomedical data in order to optimize clinical decision making, and—at the same time—minimize the amount of unnecessary examinations.

## 1. Introduction

Cardiovascular diseases (CVDs) are reportedly the major cause of death worldwide, taking an estimated 17.9 million lives each year, according to the World Health Organization [1]. Obesity, smoking, physical inactivity and high blood pressure are among the most important risk factors [2]. Even if they may be mitigated by consistent sport practice [3,4], CVDs can still be considered an actual danger for individuals engaged in competitions or strenuous training [5,6,7,8] because of intense and repeated efforts. Therefore, athletes are routinely monitored by sport physicians, who collect some biomedical and personal data and screen them by means of electrocardiography (ECG).

According to the outcome of the ECG examination, individuals are diagnosed as either at risk (positive, P) or not (negative, N). Subjects at risk are denied medical clearance for sport practice and eventually undergo further examination. The two classes are usually imbalanced, since the N class contains the majority of individuals, while the more interesting P class is under-represented. In general, a missing P (false negative, FN) may have a very high cost—in some cases the loss of a human life—while a false alarm (false positive, FP) usually has the cost of some further clinical investigations and temporary suspension of sport activities.

Classification is a *machine learning* technique which can be applied to predict categorical binary values, like P or N, and for such reason it may be of great value in the medical field, and in diagnostics in particular. Machine learning, a branch of artificial intelligence, consists in the application of computer algorithms in order to (semi) automatically extract knowledge from collected data. When classification is applied to large datasets, we usually speak of *data mining*, which is defined as “the process of discovering patterns in data (...) The patterns discovered must be meaningful in that they lead to some advantage, usually an economic advantage” [9]. As the amount of collected data has increased, researchers and physicians are interested in evaluating their diagnostic value by means of data mining, and eventually suggest that the observed variables may be changed or increased in order to support medical decisions. Several data mining methods have already been used as decision support systems for medical diagnosis [10]. These methods may be applied to large datasets in order to estimate health risk. Wong et al. [11] used Bayesian networks for early disease outbreak detection and obtained good performances on real data from an emergency department database, containing 7 years of medical data. The aim of the study was not diagnostic, since the data were collected from hospital patients who were actually ill, but rather epidemiological: to verify an incipient influenza outbreak (this approach could be adopted also in counter-terrorism, to detect a biological attack).

Campbell and Bennet [12] adopted a kernel-based method, which performed well on a medical dataset in identifying a rare disease. Still, the dataset size was limited and the proportion of interesting instances in the test set was very high (27 normal observations and 67 anomalies), compared to the prevalence of the disease in the general population.

Marinić et al. [13] adopted WEKA as a data mining tool. They applied a random forest classifier on a relatively small sample (*n* = 102) of psychiatric patients in order to diagnose Post Traumatic Stress Disorder (PTSD) and achieved significant results. Class distribution though was perfectly balanced (51 P and 51 N) and therefore different from that of the general population, where PTSD has a much lower prevalence. In addition, Fontaine et al. [14] explored data mining techniques in order to improve clinical evaluation of patients with neuropsychiatric disorders.

A data mining approach was proposed by Salam and McGrath [15] in dermatology. A multi-disease classifier improved medical diagnosis of skin disorders. Sacchi et al. [16] adopted a Naïve Bayes classification algorithm for the prediction of glaucoma. Having a small and imbalanced dataset, they applied both bootstrapping and resampling to train the model. Chan et al. [17] showed that machine learning classifiers outperformed traditional statistical approaches in the diagnosis of the same medical condition.

An increasing interest in the adoption of data mining for classification and prediction in cardiology was reported by Kadi et al. [18] in a recent systematic literature review. For example, Karaolis et al. [19] adopted a decision tree (DT) for the assessment of coronary risk.

Comparing different classification methods in medical statistics has been suggested by several authors [20,21,22] in order to assess the real advantages of one technique over the other. Still, there is a lack of studies applied to large datasets in domains where diseases have a low prevalence (like CVDs in sport medicine) but individuals may be at greater risk because of increased stress or pressure. Further, there is an on-going debate on the necessity of a sustainable and cost-effective health care [23], also by means of a more sensible use of medical tests [24].

The aims of this study were to assess the performances of a data mining method in the prediction of ECG outcomes in an imbalanced dataset, when a resampling technique is applied, and to verify whether the results may be used to support clinical decision making. The underlying hypothesis was that, given a limited set of predictive biomedical variables, data mining could achieve good predictive accuracy if the proper algorithms were trained with a large amount of data.

## 2. Materials and Methods

### 2.1. Sample

A dataset including medical examinations of 26,002 athletes, both sexes, was collected at the Polyclinic for Occupational Health and Sports in Zagreb (Croatia) by medical staff in 2012. All individuals were involved in competitive sport practice, for which medical clearance was needed. The following data were collected for all subjects: sex, age, height, weight, resting pulse rate, diastolic and systolic pressure, and ECG at rest (P or N). The largest majority (91.2%) of outcomes was N, while a minority (8.8%) was P. 

This study is the result of the collaborative research project “Health status and life quality of athletes”, involving the Institute for Anthropological Research in Zagreb, the Department of Biomedical Sciences and Surgical Specialties of the University of Ferrara, the Polyclinic for Occupational Health and Sports of Zagreb Sports Association with Laboratory of Medical Biochemistry in Zagreb, and the Interdisciplinary Center for Network Science and Applications of the University of Notre Dame. The research was approved by the Ethical Committee of the Institute for Anthropological Research in Zagreb (registration number: 1.14-1169/13).

### 2.2. Machine Learning Background

Two classification techniques were trained and tested in order to predict the class (P or N) of the athletes. DT was chosen because it allows us to describe the extracted knowledge (patterns) in a simple and intuitive way, which can be easily understood by domain experts, like medical doctors. It is commonly preferred when explanation (understanding) is as important as prediction (knowing). Further, DT is a well-established support tool in medical decision making [25,26,27,28,29].

Logistic regression (LR) is a technique commonly applied to medical datasets, half-way between classic statistics and machine learning. The main difference between regression and classification is that in the former the predicted variable is numeric, while in the latter it is categorical. Since in logistic regression, the predicted variable can have only two numeric values (1 or 0), it can be used for binary classification [30], where 1 stands for P and 0 for N.

The assessment of classification performance is a major issue in imbalanced datasets. Usually, the basic evaluation index is accuracy, the rate of correct guesses (i.e., true positives, TP, and true negatives, TN) on total instances (TP + TN)/(P + N). It is an acceptable choice when class distribution is symmetric or close to it.

In case the distribution is imbalanced, accuracy can be misleading, unless a trivial solution is acceptable [31]. In fact, given a low prevalence, a high accuracy can be easily achieved classifying all instances as N, but it would imply missing all P individuals. Therefore, classification algorithms may have a high specificity or true negative rate (TNR = TN/N), as the majority class (N) is well represented, while sensitivity, or true positive rate (TPR = TP/P) may be significantly lower, as the minority class (P) is under-represented. Therefore, performance indexes other than accuracy should be taken into consideration.

A trade-off between TPR and TNR can be represented in receiver operating characteristic (ROC) space (for an introduction see [32]). Different cut-off values for the same classifier correspond to points in ROC space. The interpolation of such points draws a curve. The area under the ROC curve (AUC) is the accepted standard in the assessment of classification performances in imbalanced datasets [33], in particular in diagnostic systems [34,35,36].

Youden index J = TPR + TNR − 1 [37] is a summary measure corresponding to the distance between a point (corresponding to a cut-off value) on the ROC curve and the underlying 45° (random-guessing) line. J represents the probability of an informed decision. It has been used to assess the ability of biomarkers to correctly classify healthy and non-healthy individuals when equal weight is given to both sensitivity and specificity [38,39].

Different resampling methods [40] can be applied in order to balance the class distribution and thus improve performances. Resampling can either undersample the majority class and/or oversample the minority class. Synthetic Minority Oversampling Technique (SMOTE) [41] has proved to be reliable in different domains, including the prediction of type 2 diabetes [42], SMOTE does not simply duplicate existing instances, which would easily lead to overfitting. It creates a new instance in feature (i.e., variable, in data mining jargon) space, between an existing instance and one, randomly-chosen, of its k nearest neighbors. Euclidean distance between two neighboring instances is calculated and then it is multiplied by a random number between 0 and 1. The distance is used to calculate the position in feature space of the new instance.

### 2.3. Data Cleaning and Resampling

The raw dataset was cleaned, removing outliers. Body weight and height were replaced by body mass index (BMI = weight/height^2^), a common proxy for obesity and cardiovascular risk in the general population [43,44,45,46,47]. Only systolic pressure was used, since systolic and diastolic blood pressures are correlated, and according to recent findings, the former is a better predictor of risk [48,49].

SMOTE was applied to the minority class with 100% oversampling (P instances were doubled). Higher percentages were tried but they did not improve classification or led to overfitting. Then random undersampling was applied to the majority class with an even distribution spread, in order to balance the two classes, so that class ratio P/N was set to 1.

### 2.4. Data Mining and Statistical Analyses

The dataset was divided into training (66% of total instances) and test (remaining 34%). DT was first applied without resampling (1st run), then with resampling (2nd run, k = 5). In the latter case, only training set was resampled, while test set kept the same distribution as the original subset (which is supposed to be similar to the population, giving the high cardinality of the sample). This prevented performances from being artificially high. Pulse rate was the best predictor in both cases. Two thresholds were identified: low (L) and high (H). Risk was low in between, moderate for pulse rate <L and high for pulse rate >H.

LR was applied twice. In the 1st run it was applied to all the collected variables directly. In the 2nd run, a discrete variable was created, with the following values: 0 (low risk) for pulse rate values between L and H, 1 (moderate risk) for values <L and 2 (high risk) for values >H. This variable replaced pulse rate in a data-driven way, instead of using a-priori values from the literature. Cross-validation (10-fold) was used to assess the model’s performances and test whether this statistical regression technique could improve the performances of a standard machine learning algorithm like DT.

TPR, TNR, J and AUC were used to assess the performances of the adopted method, which is described in Figure 1.

MS Excel 2016 (Microsoft Corporation, Redmond, WA, USA) was used for data collection and data cleaning. WEKA 3.6 (University of Waikato, Waikato, New Zealand) was used for data mining. Stata IC 13.1 (StataCorp LLC, College Station, TX, USA) was used to perform LR, cross validation and ROC analysis.

## 3. Results

Descriptive statistics by sex and age classes are shown in Table 1. Values are reported as mean and standard deviation, with the exception of ECG positives, which are reported as count and percentage.

Classification performances of DT and LR are shown in Table 2.

DT performances improved considerably by means of resampling, particularly in terms of greater sensitivity, from 0.29 to 0.68.

LR was highly significant in both runs (*p* < 0.001). Still, in the first run, with the default cut-off = 0.5, performance was so low (AUC = 0.56) that no further attempt with other cut-off values was made. DT performed better, even without resampling.

After a discrete variable was introduced in place of pulse rate, LR increased sensitivity (with cut-off = 0.12) from 0 to 0.65, without considerably diminishing specificity. This result was achieved because risk was high in both the lower and upper range of the continuous variable, and not monotonously increasing. It became evident after DT was applied, since it classified as P instances with either low or high pulse rate values. L and H were found inductively in the data (L = 60 bpm, H = 99 bpm), but they were close to those which can be found in domain-specific literature.

## 4. Discussion

In the present study, we tested the effectiveness and accuracy of a data mining methodology in the prediction of cardiovascular risk in athletes. The application of resampling and two classification techniques on an imbalanced dataset was evaluated. Resampling improved classification by means of DT. LR has proved to be not an accurate diagnostic tool if applied directly to continuous variables, since in the medical domain high risk is often associated with feature values which are either too low or too high. This is particularly true in the assessment of cardiovascular risk, since pulse rate and blood pressure have upper and lower thresholds.

Feature cut-off values, which were acquired inductively from the collected data, can be immediately converted into a small set of simple decision rules to assist the diagnostic process. Additionally, they can be used to create categorical or discrete variables for LR, in order to improve its performances, without the need for a-priori assumptions. The chosen predictors can be easily measured by a family practitioner at almost no cost, which would make cardiovascular risk evaluation very efficient.

Classification performances were not assessed by means of accuracy because of the asymmetric distribution of the two classes. AUC and J have been suggested, because they represent a trade-off between sensitivity and specificity, in case they are both given the same importance. Even if this assumption can be reasonable from a statistical point of view, in medical diagnostics it poses a serious ethical concern. In fact, a FP comes at a cost of some unnecessary examinations and temporary suspension of sport activities, while a FN may imply the loss of a human life, in case a serious condition is present. Since a FN has a higher cost than a FP, a cost matrix may be adopted, where correct predictions (TP and TN) have no cost, and errors (FN and FP) have different weights [50]. Still, in the medical domain, it is not always possible to compare the cost of errors. Since there is no general agreement on the amount of resources which can be allocated to reduce the risk related to FN, it is not easy to give the proper weights to the two kinds of errors.

Even if resampling improved DT performances, in terms of both TPR and J, sensitivity remained lower than specificity, implying a high false negative rate. This may not be acceptable in a medical domain where a high risk may imply the loss of a human life. Increased oversampling may lead to overfitting, even if a minimum TPR may be required or imposed by medical authorities. Therefore, LR cut-off values may be lowered to improve sensitivity, even if it implies a large reduction in both specificity and J. Nonetheless, it is important to note that DT and LR performances were similar, and therefore DT alone can be adopted as a predictive algorithm, with the advantage over LR of improved understanding. DT confirmed to be a good means to capture and represent domain-specific (medical, in this case) knowledge, in an intuitive and easy-to-understand way, something which cannot be achieved by means of LR. These facts may even question the application to medical datasets of statistical techniques instead of machine learning, when a binary classification is required, as in diagnostics.

The main limitation of this study lies in the lack of comparison with a standard control method (beside ECG, which is used as a gold-standard) for cardiovascular risk assessment. In order to overcome this limitation, cardiologists and general practitioners as well should be involved in a future study from the design phase.

## 5. Conclusions

This study tested the effectiveness of a simple resampling technique in improving the assessment of cardiovascular risk, when data are imbalanced, as it is often the case in real-world situations. SMOTE improved the performances of DT in terms of greater AUC and sensitivity. In addition, DT produced actionable knowledge, which can be applied in the prediction of CVDs, diminishing the need for assumptions.

Further research is required to test the performances of the proposed approach in the general (i.e., non-athletic and older) population, to determine an acceptable and agreed classification performance index—which will highlight the best trade-off between medical risk and sustainable welfare—and to verify whether this data mining methodology can be applied to improve diagnosis and optimize healthcare policies, thanks to a reduction of unnecessary examinations.

## Figures and Tables

**Figure 1 ijerph-17-07923-f001:**
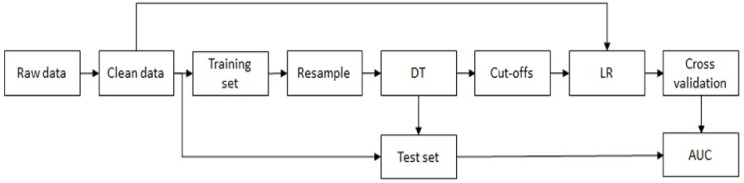
Data mining process. DT: decision tree; LR: logistic regression; AUC: area under curve.

**Table 1 ijerph-17-07923-t001:** Descriptive statistics by sex and age classes.

Variables	6–10 Years	11–14 Years	15–18 Years	≥19 Years	Total
Females	*n* = 1372	*n* = 1884	*n* = 970	*n* = 757	*n* = 4983
Weight (kg)	34.2 ± 8.8	51.7 ± 10.8	61.8 ± 8.8	65.1 ± 10.5	50.9 ± 15.1
Height (cm)	138.3 ± 10.0	160.6 ± 8.7	168.5 ± 7.0	168.9 ± 7.52	157.3 ± 14.9
BMI (kg/m^2^)	17.7 ± 2.8	19.9 ± 3.1	21.8 ± 2.6	22.8 ± 3.2	20.1 ± 3.5
Pulse rate (*n*)	83.1 ± 13.1	77.1 ± 13.0	68.3 ± 11.1	65.4 ± 11.7	75.2 ± 14.1
Systolic pressure (mm Hg)	97.3 ± 9.7	105.8 ± 9.9	108.7 ± 10.4	113.2 ± 11.8	105.2 ± 11.6
Diastolic pressure (mm Hg)	62.8 ± 7.47	66.4 ± 7.8	68.6 ± 7.8	73.3 ± 8.4	66.9 ± 8.6
ECG Ps (*n* (%))	140 (10.2)	160 (8.5)	63 (6.5)	65 (8.6)	428 (8.6)
Males	*n* = 4787	*n* = 5776	*n* = 4253	*n* = 6203	*n* = 21,019
Weight (kg)	33.6 ± 8.6	52.4 ± 13.4	71.4 ± 11.7	83.9 ± 12.7	61.3 ± 22.6
Height (cm)	136.9 ± 9.2	161.4 ± 11.6	178.6 ± 7.6	180.0 ± 7.3	164.8 ± 19.2
BMI (kg/m^2^)	17.7 ± 2.8	19.9 ± 3.3	22.3 ± 3.0	25.9 ± 3.4	21.6 ± 4.4
Pulse rate (*n*)	79.0 ± 12.6	73.1 ± 12.6	66.8 ± 12.4	63.8 ± 11.9	70.4 ± 13.7
Systolic pressure (mm Hg)	97.1 ± 9.1	106.8 ± 11.1	118.2 ± 10.9	126.4 ± 11.8	112.7 ± 15.6
Diastolic pressure (mm Hg)	61.5 ± 7.8	65.2 ± 7.8	69.6 ± 8.2	77.9 ± 9.4	69.0 ± 10.5
ECG Ps (*n* (%))	379 (7.9)	405 (7.0)	397 (9.3)	699 (11.3)	1879 (8.9)

BMI: body mass index; ECG Ps: electrocardiography positives.

**Table 2 ijerph-17-07923-t002:** Algorithm classification performances.

Algorithm	TPR	TNR	J	AUC
DT (1st run)	0.29	0.97	0.26	0.68
LR (1st run)	0	1	0.00	0.56
DT (2nd run)	0.68	0.82	0.50	0.76
LR (2nd run)	0.65	0.82	0.47	0.78

TPR: True positive rate; TNR: True negative rate; J: Youden index; AUC: Area under the ROC curve; DT: Decision tree; LR: Logistic regression.

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
