# Peer review of "Predicting Cardiovascular Risk in Athletes: Resampling Improves Classification Performance"

_ijerph, 2020, doi:10.3390/ijerph17217923_

Round 1
Reviewer 1 Report
I thank the authors for this paper. Currently, most clinical studies suffer from sampling error. This study is addressing the issue and provides evidence to improve sampling in order to make better clinical decision. My comments are as below,
Minor:
- Line 34 - Please mention the number of worldwide cases every year and the costs associated with it.
- Move table 1 to the results section
Major:
- The method section is very hard to follow and needs to be totally rewritten by explaining exactly what was done in the study, what software was used, what coding script was used, what settings were applied.
- Move background information about sampling to the introduction section and would suggest to limit the method section to explain the methods followed in the study itself.
- Expand the results on how many patients had CVD, and how the model predicted it, and report the accuracy of the DT compared to the actual outcomes.
Author Response
I thank the authors for this paper. Currently, most clinical studies suffer from sampling error. This study is addressing the issue and provides evidence to improve sampling in order to make better clinical decision. My comments are as below,
Authors: Thank you for your careful review of this paper. Your comments have been very useful to improve the manuscript.
Minor:
- Line 34 – Please mention the number of worldwide cases every year and the costs associated with it.
Authors: We added the number of deaths every year (lines 33-34).
- Move table 1 to the results section
Authors: Table 1 was moved to results section.
Major:
- The method section is very hard to follow and needs to be totally rewritten by explaining exactly what was done in the study, what software was used, what coding script was used, what settings were applied.
Authors: Thanks for your comments. The introduction and the methods sections were totally rewritten as per your suggestions. Software details were added at the end of the section 2.3.
- Move background information about sampling to the introduction section and would suggest to limit the method section to explain the methods followed in the study itself.
Authors: The method section has been reduced to contain only the adopted methods. Background information was moved to the introduction, according to your suggestions.
- Expand the results on how many patients had CVD, and how the model predicted it, and report the accuracy of the DT compared to the actual outcomes.
Authors: We have expanded the results, adding the number of patients with CVD (i.e. with ECG positive) in Table 1, separately in males and females and for each age group. Given the highly skewed distribution, the accuracy of the model has been assessed by means of sensitivity and specificity, and Youden index as a summary statistics (Table 2).
Reviewer 2 Report
The authors have done a very sophisticated statistical diagnosis of ECG results in athlete screening. What I am having difficulty in ascertaining is the precise definition of a true positive, as well as the actual result of their ROC curve analysis. Precisely what information content from the Data other than the ECG itself is extracted in this analysis?
Without explicit results of the way in which the information obtained from athletes screening is used, it is very difficult for readers to understand precisely what the outcomes being assessed are, and how the various data elements contribute to these outcomes
Author Response
Comments and Suggestions for Authors
R2: The authors have done a very sophisticated statistical diagnosis of ECG results in athlete screening. What I am having difficulty in ascertaining is the precise definition of a true positive, as well as the actual result of their ROC curve analysis. Precisely what information content from the Data other than the ECG itself is extracted in this analysis?
Authors: Thank you for your comments to improve our manuscript. The positive instances are the subjects who were diagnosed as such (P) according to ECG. Therefore, True Positives are the individuals correctly classified as such by our methods. The area under the ROC curve increased significantly from the first to the second execution, implying better sensitivity and specificity, and showing the positive effect of resampling. Beside ECG (the predicted class), our method extracted from the collected data the lower and upper thresholds for pulse rate, which were used to improve prediction. The Results section was expanded in order to facilitate understanding for the readers (lines 242-250).
R2: Without explicit results of the way in which the information obtained from athletes screening is used, it is very difficult for readers to understand precisely what the outcomes being assessed are, and how the various data elements contribute to these outcomes
Authors: Collected data were used to create a predictive model, based in one case on decision trees and in the other on logistic regression. The dependent variable (to be predicted) was ECG result (P or N), while the other biomedical variables were used as predictors to feed the model.
Reviewer 3 Report
Dear Authors,
The topic is interesting but more information should be reported to explain how much the cardivascular risk can be predicted, which are the percentage, why the data were presented in the groups 6-10, 11-14, 15-18, >19 years.....
Do not repeat in keywords words that are already in the title.
All tables and figure must be well undertood, so the meaning of abbreviations need to be added. Only table 2 is well done.
Best Regards
Author Response
Dear Authors,
The topic is interesting but more information should be reported to explain how much the cardiovascular risk can be predicted, which are the percentage, why the data were presented in the groups 6-10, 11-14, 15-18, >19 years.....
Authors: Thank you for your comments to improve our manuscript.
We added more information about the percentage of CVD in our sample (data were added in table 1, separately in males and in females and for each age group). The subdivision in 4 age groups takes into account the morphometric and functional changes that occur in the transition from childhood, pre-adolescence, adolescence, adulthood.
R3: Do not repeat in keywords words that are already in the title.
Authors: We replaced the keyword resampling (already present in title) with medical diagnostic.
R3: All tables and figure must be well understood, so the meaning of abbreviations need to be added. Only table 2 is well done.
Authors: We added the meaning of abbreviations in tables, if absent.
Round 2
Reviewer 1 Report
I thank the authors for making relevant changes. However, the introduction is tool long. Could I suggest to either add the machine learning background explanations as a supplementary or as a separate heading within the method section. If adding to the methods, please make sure it is a separate subheading and not mixed along with the actual study methods.
Author Response
R: I thank the authors for making relevant changes. However, the introduction is tool long. Could I suggest to either add the machine learning background explanations as a supplementary or as a separate heading within the method section. If adding to the methods, please make sure it is a separate subheading and not mixed along with the actual study methods.
Authors: Thank you for your actual help to improve our manuscript. As you suggested, we added the machine learning background as a separate heading within the method section.
